# Decorin-mediated suppression of tumorigenesis, invasion, and metastasis in inflammatory breast cancer

Xiaoding Hu[1,2], Emilly S. Villodre [1,2], Richard Larson[2,3], Omar M. Rahal[2,3], Xiaoping Wang[1,2], Yun Gong[2,4], Juhee Song[5], Savitri Krishnamurthy[2,4], Naoto T. Ueno [1,2], Debu Tripathy [1,2], Wendy A. Woodward[2,3] & Bisrat G. Debeb [1,2 ✉]

Inflammatory breast cancer (IBC) is a clinically distinct and highly aggressive form of breast cancer with rapid onset and a strong propensity to metastasize. The molecular mechanisms underlying the aggressiveness and metastatic propensity of IBC are largely unknown. Herein, we report that decorin (DCN), a small leucine-rich extracellular matrix proteoglycan, is downregulated in tumors from patients with IBC. Overexpression of DCN in IBC cells markedly decreased migration, invasion, and cancer stem cells in vitro and inhibited tumor growth and metastasis in IBC xenograft mouse models. Mechanistically, DCN functioned as a suppressor of invasion and tumor growth in IBC by destabilizing E-cadherin and inhibiting EGFR/ERK signaling. DCN physically binds E-cadherin in IBC cells and accelerates its degradation through an autophagy-linked lysosomal pathway. We established that DCN inhibits tumorigenesis and metastasis in IBC cells by negatively regulating the E-cadherin/EGFR/ERK axis. Our findings offer a potential therapeutic strategy for IBC, and provide a novel mechanism for IBC pathobiology.

[1] Department of Breast Medical Oncology, the University of Texas MD Anderson Cancer Center, Houston, TX 77030, USA. [2] Morgan Welch Inflammatory Breast Cancer Clinic and Research Program, the University of Texas MD Anderson Cancer Center, Houston, TX 77030, USA. [3] Department of Radiation Oncology, the University of Texas MD Anderson Cancer Center, Houston, TX 77030, USA. [4] Department of Pathology, the University of Texas MD Anderson Cancer Center, Houston, TX 77030, USA. [5] Department of Biostatistics, the University of Texas MD Anderson Cancer Center, Houston, TX 77030, USA. ✉email: bgdebeb@mdanderson.org

Inflammatory breast cancer (IBC) is an aggressive, clinically and pathologically distinct form of locally advanced breast cancer. IBC is particularly fast growing, invasive, and metastatic. Nearly all women have lymph node involvement at the time of diagnosis, and ~36% have gross distant metastases[1,2]. Although IBC is considered rare, constituting only 2–4% of breast cancer cases in the United States, it accounts for a disproportionate 10% of breast cancer-related deaths annually[3]. The prognosis for patients with IBC remains poor with a 5-year disease-free survival rate of only 40% despite multimodality treatment[4]. This underscores the critical need to better define the mechanisms that dictate the aggressive behavior of IBC and to develop new therapeutic targets and novel agents to improve the overall prognosis for women with IBC.

Major efforts have been undertaken to elucidate IBC tumor biology and to identify molecular alterations distinct to IBC that potentially might be translated into novel therapeutic strategies. Several important targets and pathways have been identified, including overexpression of EGFR[5], overexpression of IFITM1[6], RhoC[7], E-cadherin[8], XIAP[9], TIG1 and Axl[10], and eIFG4I[11], and downregulation of TGFβ and WISP3[7], as well as pathways associated with enrichment of the stem cell phenotype[12,13] and angiogenesis[8]. These efforts have significantly contributed to our understanding of IBC; however, the molecular basis for the unique and aggressive biology of IBC is still not well understood, and effective targeted therapies for this disease remain limited.

The overexpression of E-cadherin is a notable finding that distinguishes IBC from other breast cancers. E-cadherin expression is indicative of low metastatic potential in most cancers. Loss of its expression contributes to increased proliferation, invasion, and metastasis in breast cancer[14,15]. In IBC, despite its enhanced aggressive and metastatic behavior, E-cadherin is overexpressed in tumor cells, tumor emboli, and metastases as well as in IBC cell lines[3,8,16–18]. The presence of E-cadherin augments invasion and tumorigenesis in preclinical in vitro and in vivo IBC models[19–21], and supports the formation of tumor emboli, a hallmark of IBC[17,21]. These findings strongly support a distinct, oncogenic role for E-cadherin in IBC tumors. However, it is not known how E-cadherin is regulated in IBC and which pathways are associated with E-cadherin-mediated IBC tumorigenesis and metastasis.

Decorin (DCN) is an extracellular matrix protein that belongs to the small leucine-rich proteoglycan family. It regulates a vast array of cellular processes including collagen fibrillogenesis[22], wound repair[23], angiostasis[24], tumor growth[25], and autophagy[26,27]. DCN acts as a potent suppressor of tumor growth in many solid tumors including breast cancer[28–30]. Further reinforcing the oncosuppressive role of DCN, systemic delivery of the DCN protein core or adenoviral transduction of DCN attenuated primary tumor growth and metastasis in breast cancer cells[29,30]. However, the functional role of DCN in IBC aggressiveness and tumorigenesis remains to be explored. DCN is known to bind and antagonize various receptor tyrosine kinases and autocrine factors to inhibit downstream oncogenic signaling, thereby blocking the growth of cancer cells and tumor xenografts[31–35]. One of DCN's key interacting partners is EGFR, a known mediator of tumorigenesis and metastasis in IBC mouse models and an important therapeutic target in IBC patients[36–38]. EGFR is an independent predictor of poor prognosis and increased recurrence in IBC patients, and a promoter of invasion and metastasis in preclinical mouse models[36,37]. Importantly, in a phase II clinical trial, a humanized anti-EGFR antibody panitumumab in combination with neoadjuvant chemotherapy led to remarkably high pathological complete response rate in patients with triple-negative IBC[38], indicating that EGFR is a promising therapeutic target in IBC.

In the present study, we found that DCN alters the E-cadherin–EGFR–ERK axis to inhibit invasion and tumor growth

of IBC cells. We also found that DCN interacts with and accelerates the degradation of E-cadherin via an autophagy-linked lysosomal pathway.

## Results

**DCN inhibits in vitro tumorigenic features in IBC cell lines.** Previous research has shown DCN to be a tumor suppressor in different tumor types, including breast cancer[28–34]. Consistent with these findings, our analysis of publicly available datasets showed that DCN mRNA levels were downregulated in several malignant cancers (Supplementary Fig. 1a). DCN expression was also significantly lower in breast tumors than in normal breast tissues (Supplementary Fig. 1b, c), and in the more aggressive basal-like breast cancer subtype compared with the luminal subtype (Supplementary Fig. 1d). Moreover, high DCN expression in patients with breast cancer correlated with better survival outcomes (Supplementary Fig. 1e). In IBC, DCN mRNA is significantly downregulated in IBC tumors versus normal breast (Supplementary Fig. 1f). Moreover, we analyzed public database (GSE5847) and found that DCN mRNA was expressed in both cancer cells and stroma, with higher expression seen in the tumor cells (Supplementary Fig. 1g). Because all of these analyses were based upon RNA expression levels, we used immunohistochemical staining to assess DCN protein expression in a tissue microarray containing 65 samples of IBC and 22 of non-IBC, locally advanced primary breast cancer (LABC). DCN protein, localized to the cytoplasm and membrane, was expressed in fewer IBC than in LABC patient tumors (2 of 65 IBC vs. 5 of 22 LABC; $p = 0.01$; Supplementary Fig. 1h). Although these results suggest that DCN may act as a tumor suppressor in IBC, the functional role of DCN in IBC tumor aggressiveness and metastasis is unknown.

To investigate the functional role of DCN in IBC, we first generated stable DCN overexpression in four IBC cell lines [ER−/PR−/HER2+: MDA-IBC3[39], and SUM190; ER−/PR−/HER2−: SUM149 and BCX010[40]]. Overexpression of DCN protein and mRNA levels versus paired control lines was confirmed via western blotting (Fig. 1a) and real-time PCR (Supplementary Fig. 2). DCN overexpression had no effect on proliferation (Supplementary Fig. 3) but significantly inhibited colony formation in all of the DCN-overexpressing IBC cell lines (Fig. 1b). Because patients with IBC are at high risk of recurrence and metastatic disease[41] and because IBC tumors are stem cell-enriched[12], we further investigated the effect of DCN overexpression on cell migration and invasion as well as cancer stem cell features. The migration and invasion of DCN-overexpressing SUM149 and BCX010 cells was significantly reduced relative to control cells (Fig. 1c, d). Of note, MDA-IBC3 and SUM190 cells are non-migratory and non-invasive in vitro. We evaluated the cancer stem cell and self-renewal ability of DCN-overexpressing IBC cell lines by using a mammosphere assay[42]. We found that both primary and secondary mammosphere formations were decreased in all DCN-overexpressing cell lines (Fig. 1e, f). We further found that DCN overexpression inhibited spheroid-forming efficiency and reduced the average spheroid size in all four IBC cell lines (Supplementary Fig. 4a, b). Because DCN is a secreted protein[35], we next asked whether exogenous DCN affects in vitro tumorigenic features in IBC cell lines. We found that treating parental IBC cell lines with recombinant DCN protein (8 µg/mL) significantly attenuated in vitro colony formation (Fig. 1g), migration (Fig. 1h), invasion (Fig. 1i), primary mammosphere formation (Supplementary Fig. 4c), spheroid-forming efficiency (Supplementary Fig. 4d), and reduced the average spheroid size (Supplementary Fig. 4e), supporting similar effects observed in the stably

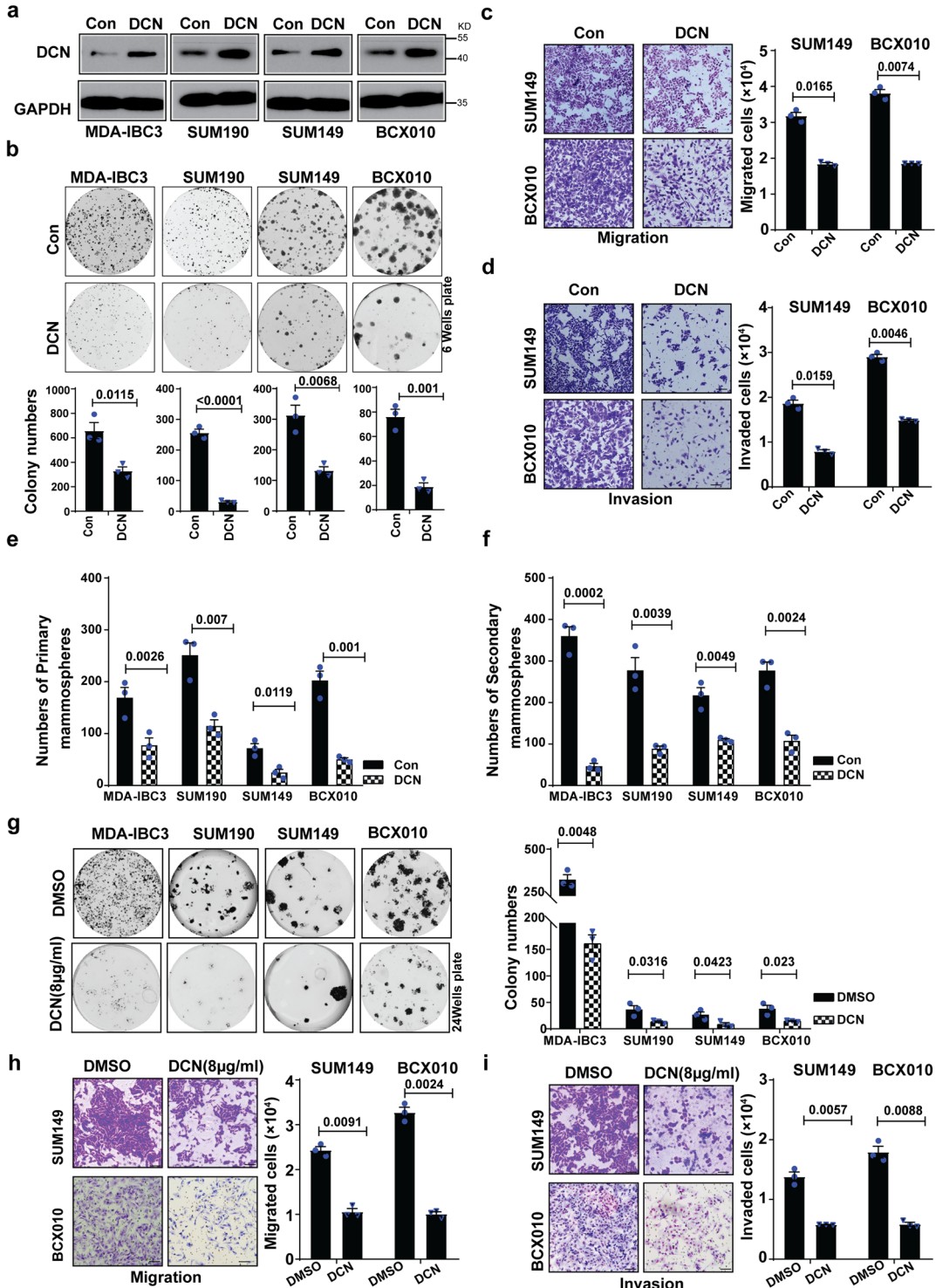

**Fig. 1 DCN inhibits in vitro tumorigenic features in IBC cell lines. a** Generation of four DCN-overexpressing IBC stable cell lines [HER2+: MDA-IBC3, SUM190; TNBC: SUM149, BCX010]. Total cell lysates were analyzed by western blotting with anti-DCN and anti-GAPDH antibody. **b–f** DCN overexpression in IBC cell lines suppresses colony formation (**b**), cell migration in SUM149 and BCX010 cells (**c**), cell invasion in SUM149 and BCX010 cells (**d**), primary mammosphere formation (**e**), and secondary mammosphere formation (**f**). **g–i** Treating IBC cell lines with recombinant DCN protein (8 μg/mL) suppresses colony formation (**g**), cell migration (**h**), and cell invasion (**i**). Scale bar: 100 μm. *P* values are from Student's unpaired *t* tests. Data are presented as mean ± s.e.m.; Data shown are representative of three independent experiments.

DCN-expressing cell lines described above. Collectively, these findings demonstrated that DCN is downregulated in IBC tumors and that it suppresses the aggressive behavior of IBC cells, including migration, invasion, and self-renewal, without affecting cell proliferation.

**DCN inhibits tumor growth and metastasis in vivo.** To investigate the function of DCN in IBC tumor growth and progression, we used orthotopic xenograft transplantation of control and DCN-overexpressing IBC cells into the cleared mammary fat pad of immunocompromised SCID/Beige mice[42]. Mice implanted

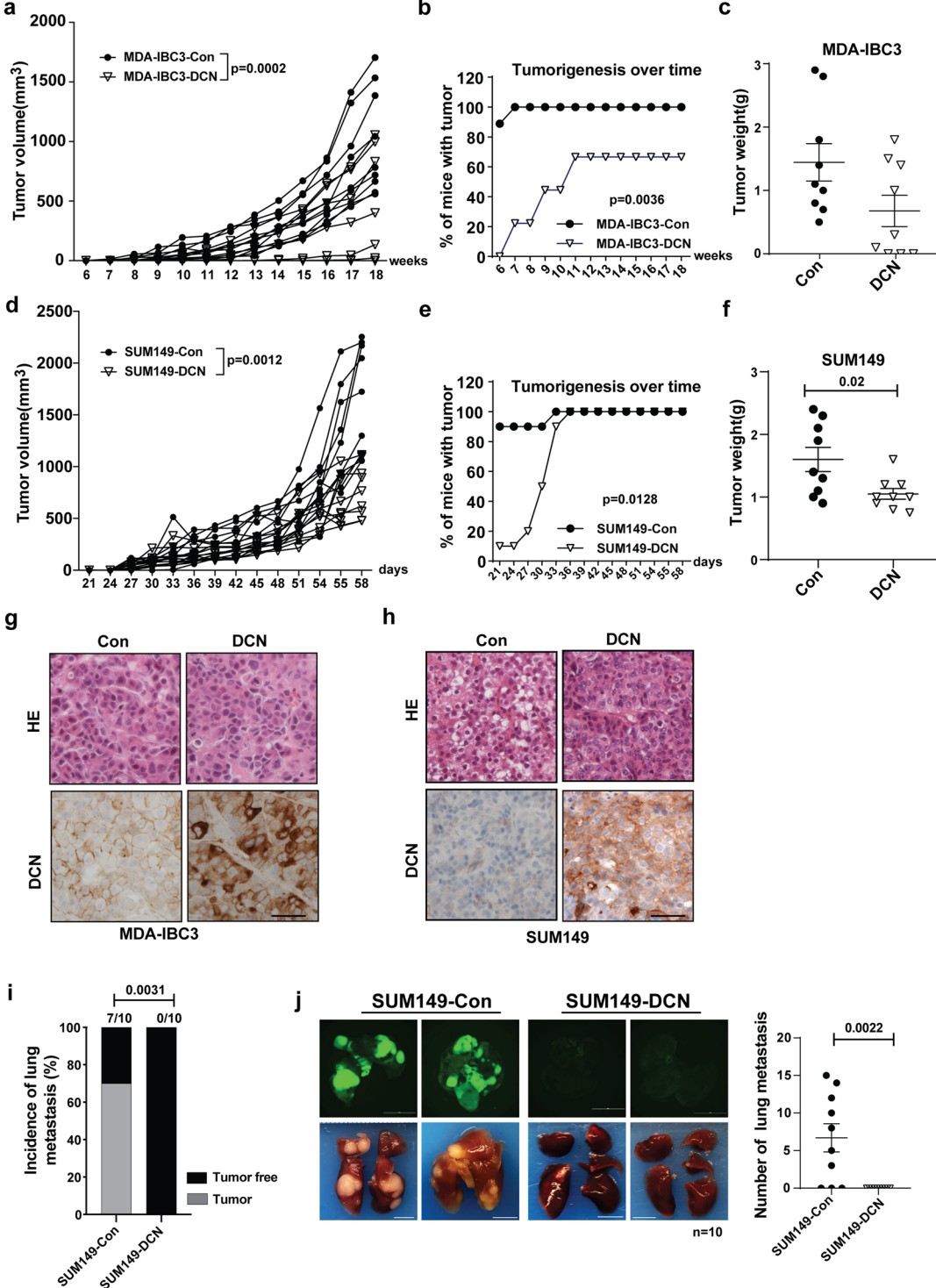

with DCN-overexpressing SUM149 or MDA-IBC3 cells had significantly reduced tumor size, tumor growth rates, and tumor weights as compared with control mice (Fig. 2a–f). Immunohistochemical staining confirmed overexpression of DCN in tumors generated from DCN-overexpressing cells (Fig. 2g, h). We also injected control or DCN-overexpressing SUM149 cells into the tail veins of SCID/Beige mice to evaluate lung metastatic colonization by incidence (number of mice with any lung metastases), average burden among mice with lung metastases, and average number of lung metastases per mouse. DCN completely inhibited the incidence of lung metastasis as compared with the control group (0% DCN vs. 70% control, $p = 0.0031$, Fisher exact test,

Fig. 2i). We also found a marked reduction in lung metastasis burden and number of lung metastasis nodules in mice injected with DCN-overexpressing cells relative to mice injected with control cells (Fig. 2j, Supplementary Fig. 5). Taken together, our findings demonstrate that DCN inhibits tumor growth and metastasis in two IBC xenograft models representing TNBC and HER2 overexpressing IBC.

**DCN reduces E-cadherin expression and EGFR pathway activation in IBC.** To elucidate the possible mechanisms through which DCN inhibits IBC invasion and tumorigenesis, we used reverse-phase protein array proteomic profiling to compare

**Fig. 2 DCN inhibits tumor growth and metastasis in IBC tumor xenograft models.** For tumor growth studies, DCN-overexpressing and control MDA-IBC3 and SUM149 cells were injected into the cleared mammary fat pads of SCID/Beige mice (9 mice/group for each cell line) to allow the formation of tumors. **a** Tumor volume is significantly decreased in DCN-overexpressing MDA-IBC3 versus control group. Data are shown as mean ± s.e.m. *P* values are from Student's unpaired *t* tests. **b** DCN-overexpressing MDA-IBC3 cells showed longer tumor latency in MDA-IBC3 xenografts (Chi-square test). **c** Tumor weight is decreased in DCN-overexpressing MDA-IBC3 tumors than in controls. *P* values from Student's unpaired *t* tests. **d** Tumor volume is significantly inhibited in DCN-overexpressing SUM149 versus control group. Data are shown as mean ± s.e.m. *P* values from Student's unpaired *t* tests. **e** DCN-overexpressing showed longer tumor latency in SUM149 xenografts (Chi-square test). **f** Tumor weight is decreased in DCN-overexpressing SUM149 tumors relative to controls. *P* values from Student's unpaired *t* tests. **g**, **h** Hematoxylin-eosin and immunostains of tumors generated from MDA-IBC3 (**g**) and SUM149 (**h**) control and DCN-overexpressing cells validates the overexpression of DCN in xenograft tumors. Scale bar: 100 μm. For lung metastatic colonization studies, GFP-labeled DCN-overexpressing and control SUM149 cells were injected via tail vein into SCID/Beige mice (10 mice/group). **i** DCN significantly inhibited the incidence of lung metastasis compared with the control group (0% DCN vs. 70% Control; *p* = 0.0031, Fisher exact test). **j** Tumor burden was significantly reduced in DCN-overexpressing mice group; scatter plot shows reduction in numbers of lung metastasis nodules in the DCN-overexpressing group (*p* = 0.0022, Wilcoxon rank-sum test was used). Scale bar: 5 mm.

control and DCN-overexpressing cells. The levels of many proteins, including E-cadherin and EGFR, were reduced in the DCN-overexpressing MDA-IBC3 and SUM149 cells relative to control cells (Supplementary Data 1). We first confirmed that DCN overexpression decreased the levels of E-cadherin and EGFR protein and suppressed EGFR pathway activation in IBC cells (Fig. 3a). We analyzed parental IBC cell lines (MDA-IBC3, SUM190, SUM149, and BCX010) treated with vehicle or recombinant DCN protein (4 or 8 μg/mL for 10 h) by western blot analysis. Treatment with DCN protein inhibited E-cadherin and EGFR levels and suppressed EGFR/ERK signaling (Fig. 3b), mirroring the results obtained from the ectopic expression of DCN. We further substantiated these findings by using immunohistochemical staining and western blot analysis of xenograft tumor samples (described in Fig. 2). The levels of E-cadherin and EGFR protein were decreased in the tumor samples from DCN-overexpressing MDA-IBC3 and SUM149 xenografts, as shown by western blotting (Fig. 3c, d) and immunohistochemical staining (Fig. 3e, f).

Given the importance of ligands for the activation of the EGFR pathway, we explored EGFR–ERK pathway activation in response to EGF ligand in DCN-overexpressing and control IBC cell lines. IBC cells grown to 70–80% confluence were serum-starved overnight and treated with EGF (50 ng/mL) for different times. We found that DCN attenuated EGF-induced phosphorylation of EGFR and ERK (Fig. 3g). Analysis of the p-EGFR/EGFR and p-ERK/ tERK ratios is shown in Supplementary Fig. 6a and b.

Further, because the loss of E-cadherin has been regarded as a crucial step in activating the epithelial–mesenchymal transition (EMT), we asked whether the significant inhibition of E-cadherin by DCN overexpression induces EMT in IBC cells. We tested the protein expression of known EMT markers, including Fibronectin, N-cadherin, Vimentin, Snail, Slug, Zeb1, and Twist, and found that DCN downregulates E-cadherin without affecting the expression of these EMT proteins (Fig. 3h). Moreover, no changes were observed in the morphology of DCN-overexpressing IBC cells as compared with controls. Our findings demonstrate that DCN-mediated inhibition of E-cadherin expression is insufficient to induce EMT in IBC cells.

**DCN suppresses aggressiveness in IBC by regulating the E-cadherin–EGFR axis.** On the basis of previous findings supporting E-cadherin as an oncogene in IBC[3,8,18–21] and our own findings supporting a possible association of DCN and inhibition of E-cadherin and EGFR activation, we speculated that DCN may suppress invasion and tumor growth in IBC via inhibition of EGFR signaling, which is dependent on E-cadherin for its regulatory function. To investigate this hypothesis and establish E-cadherin as a functional mediator of DCN biology in IBC, we first established stable E-cadherin knockdown in four IBC cell lines

(MDA-IBC3, SUM190, SUM149, and BCX010) by using two independent lentiviral shRNAs. Knockdown of E-cadherin protein and mRNA levels was confirmed via western blot analysis (Fig. 4a) and PCR (Supplementary Fig. 7). We found that knockdown of E-cadherin suppressed the phosphorylation of EGFR and ERK1/2 in all IBC cell lines, without affecting the levels of DCN protein (Fig. 4a). Next, we showed that E-cadherin knockdown in IBC cells suppressed pro-tumorigenic traits as evidenced by a significant reduction in colony formation (Fig. 4b), migration (Fig. 4c), invasion (Fig. 4d), primary (Fig. 4e) and secondary mammosphere formation efficiency (Fig. 4f), spheroid-forming efficiency (Supplementary Fig. 8a), and average size of spheroids (Supplementary Fig. 8b). Next, we asked if expression of E-cadherin in DCN-overexpressing cells rescues the tumorigenic features and the activation of EGFR/ERK pathway. To investigate this, we transfected Strep-flagged E-cadherin into DCN-overexpressing IBC cell lines and analyzed migration, invasion, and EGFR and ERK phosphorylation 48 h later. E-cadherin overexpression restored, at least partially, EGFR activation (Fig. 4g), and increased the proportion of both migrating (Fig. 4h) and invading cells (Fig. 4i). Collectively, these results suggest that the downregulation of E-cadherin mediates at least in part the DCN-mediated suppression of invasion and repression of EGFR signaling in IBC cells.

**DCN interacts with E-cadherin.** Our findings that E-cadherin depletion had the same anti-tumorigenic effects as DCN overexpression or DCN recombinant protein, and that E-cadherin rescued the effects of DCN, confirm E-cadherin as a potential functional partner of DCN in IBC cells. However, the mechanism by which DCN regulates E-cadherin is unknown. Given the physical association noted between overexpressed DCN and E-cadherin in a colorectal cancer cell line[43], we first examined whether DCN associates with E-cadherin in four IBC cell lines and HEK293T cells by reciprocal immunoprecipitation of the exogenous or endogenous proteins followed by immunoblotting analysis. We detected E-cadherin in the immune precipitates captured by anti-Flag-DCN antibody (Fig. 5a). Conversely, DCN was captured in the anti-Flag-E-cadherin immunoprecipitate (Fig. 5b). The same results were obtained in HEK293T cells (Supplementary Fig. 9). This interaction was confirmed at endogenous protein levels in HEK 293T cells by using anti-DCN and anti-E-cadherin antibodies (Fig. 5c). Our results showed a robust physical association between DCN and E-cadherin in all IBC cell lines. To delineate the regions of E-cadherin involved in the DCN−E-cadherin interaction, various truncation mutants[44] of E-cadherin were generated as shown in Fig. 5d (left panel). We performed co-immunoprecipitation assays of full-length hemagglutinin [HA]-DCN and SBP-flag-E-cadherin truncated mutants by using Streptavidin agarose (Fig. 5d, right panel) or protein

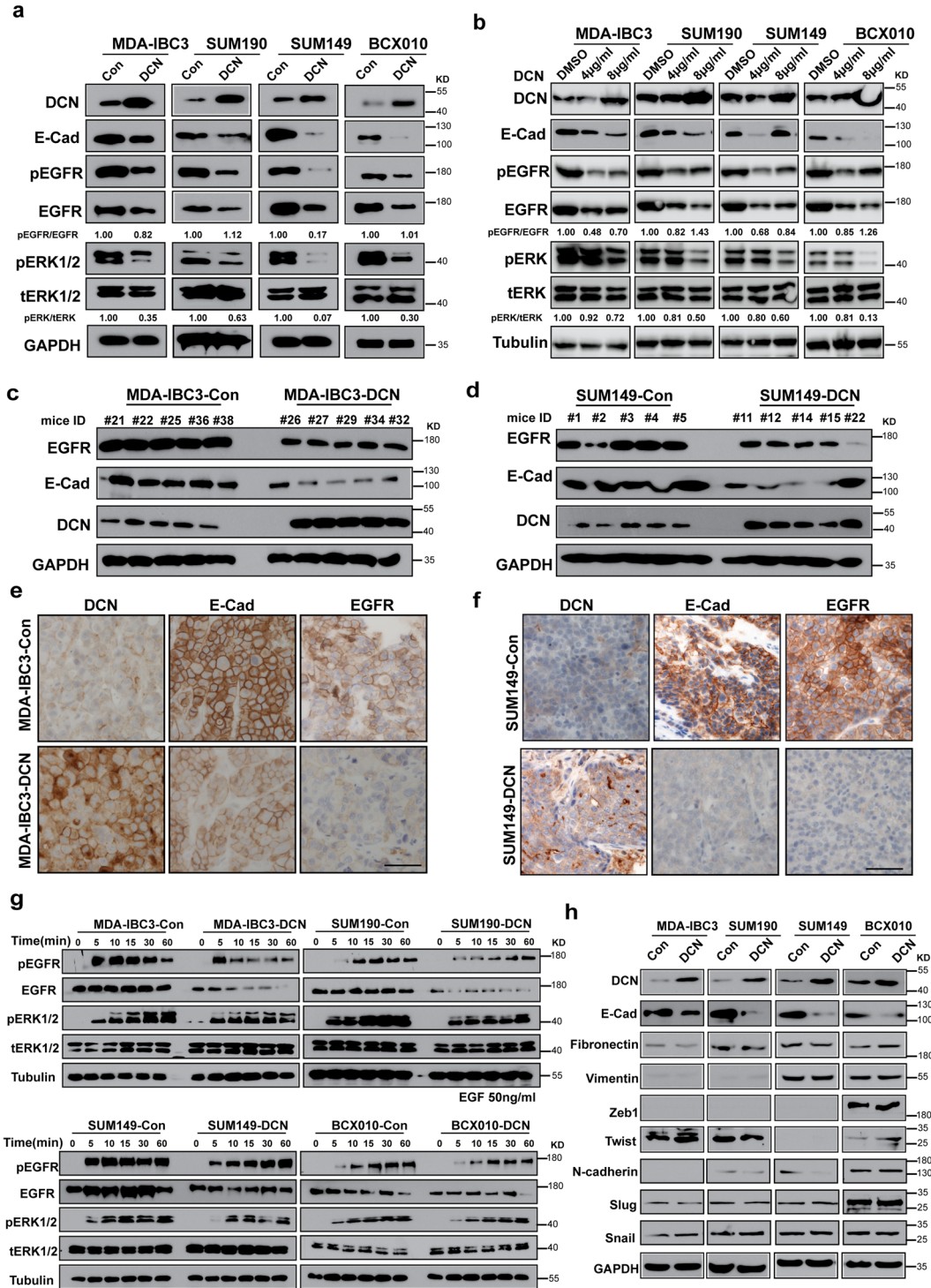

**Fig. 3 DCN inhibits E-cadherin expression and EGFR pathway activation in IBC. a** DCN suppresses E-cadherin expression and EGFR signaling in IBC cells. Expression levels of E-cadherin and EGFR are decreased in DCN-overexpressing MDA-IBC3, SUM190, SUM149, and BCX010 cells; also, the phosphorylation of EGFR (pEGFR) and ERK1/2 (pERK1/2) was suppressed in DCN-overexpressing IBC cell lines. Total ERK1/2 (tERK1/2) remains unchanged. GAPDH served as a loading control. **b** Treatment of IBC cells with DCN protein (4 or 8 µg/mL) for 2 h suppresses E-cadherin expression and EGFR pathway activation. Tubulin served as a loading control. **c** and **d** Western blot validation of E-cadherin and EGFR downregulation in tumor samples obtained from mammary fatpad transplantation of control or DCN-overexpressing MDA-IBC3 (**c**) or SUM149 (**d**) cells. **e** and **f** Immunohistochemical staining validation of E-cadherin and EGFR downregulation in tumor samples obtained from mammary fatpad transplantation of control or DCN-overexpressing MDA-IBC3 (**e**) or SUM149 (**f**) cells. Scale bar: 100 µm. **g** DCN inhibits EGFR signaling in IBC cells independently of EGF stimulation. DCN-overexpressing and control IBC cell lines were stimulated with 50 ng/mL EGF for the indicated number of hours, and total cell lysates were analyzed by western blotting. Both the total levels and the phosphorylation levels of EGFR and ERK1/2 were detected by western blotting. Tubulin served as a loading control. **h** DCN-mediated inhibition of E-cadherin does not affect expression of epithelial–mesenchymal transition markers. Cell lysates containing 40 µg of total protein were analyzed by western blotting with anti-E-cadherin, fibronectin, vimentin, and DCN antibodies. GAPDH served as a loading control.

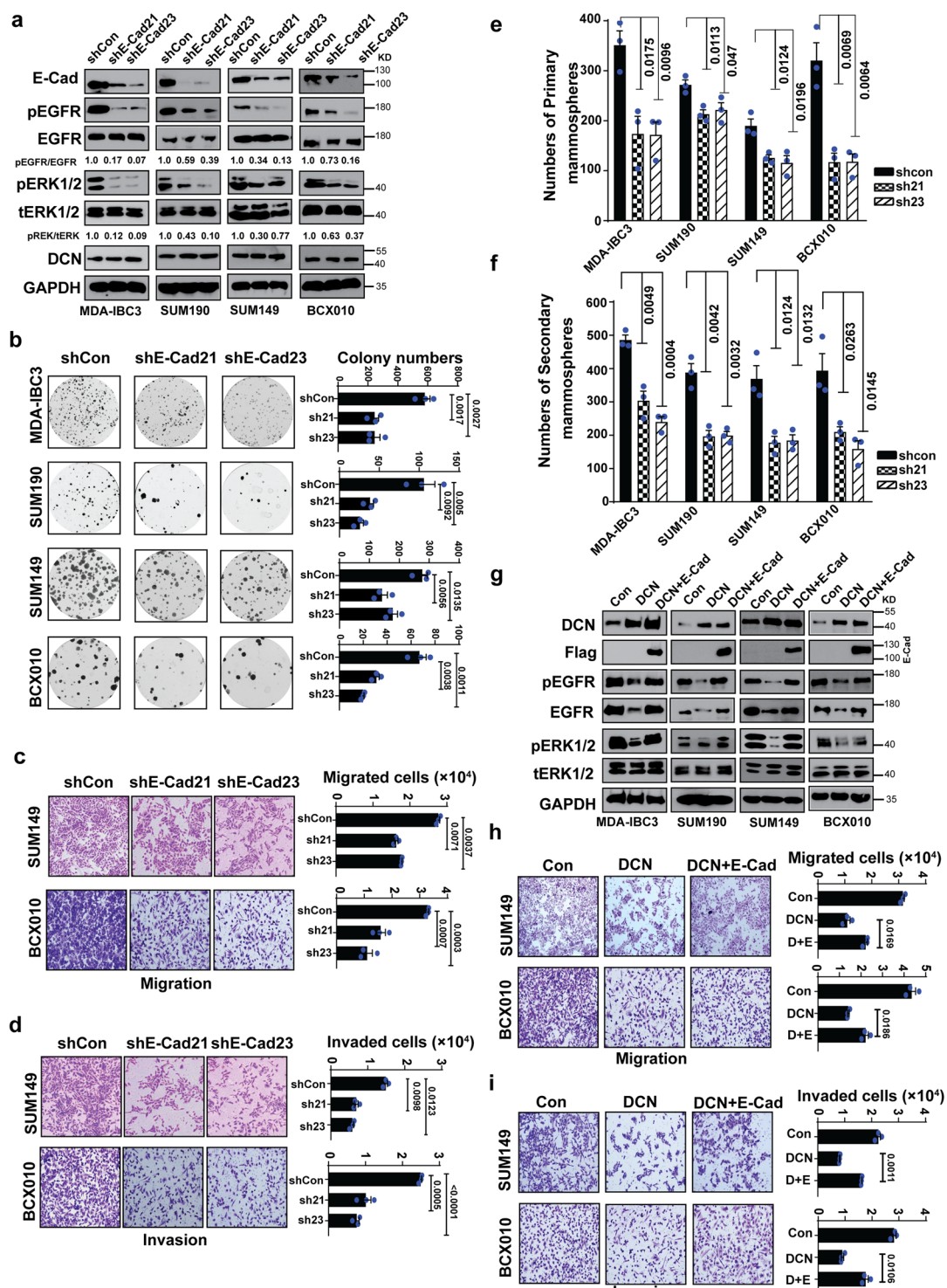

**Fig. 4 DCN suppresses aggressiveness in IBC by regulating the E-cadherin–EGFR axis. a** E-cadherin knockdown in IBC cells results in the inhibition of EGFR signaling but does not affect DCN expression. E-cadherin knockdown in IBC cells lines was achieved by transduction with two independent lentiviral shRNAs. Total cell lysates were analyzed by western blotting for EGFR pathway analysis. GAPDH served as a loading control. **b–f** E-cadherin knockdown in IBC cell lines suppresses colony formation (**b**), migration (**c**), invasion (**d**), primary mammosphere formation (**e**), and secondary mammosphere formation (**f**). **g** Restoring E-cadherin rescues EGFR signaling in DCN-overexpressing IBC cells. Flag-E-cadherin plasmid was transfected into DCN-overexpressing IBC cell lines for 48 h and cell lysates were subjected to western blotting. **h** and **i** Restoring E-cadherin increases migration (**h**) and invasion (**i**) in DCN-overexpressing SUM149 and BCX010 IBC cell lines. *P* values are from Student's unpaired *t* tests. Data are shown as mean ± s.e.m. Data shown are representative of three independent experiments.

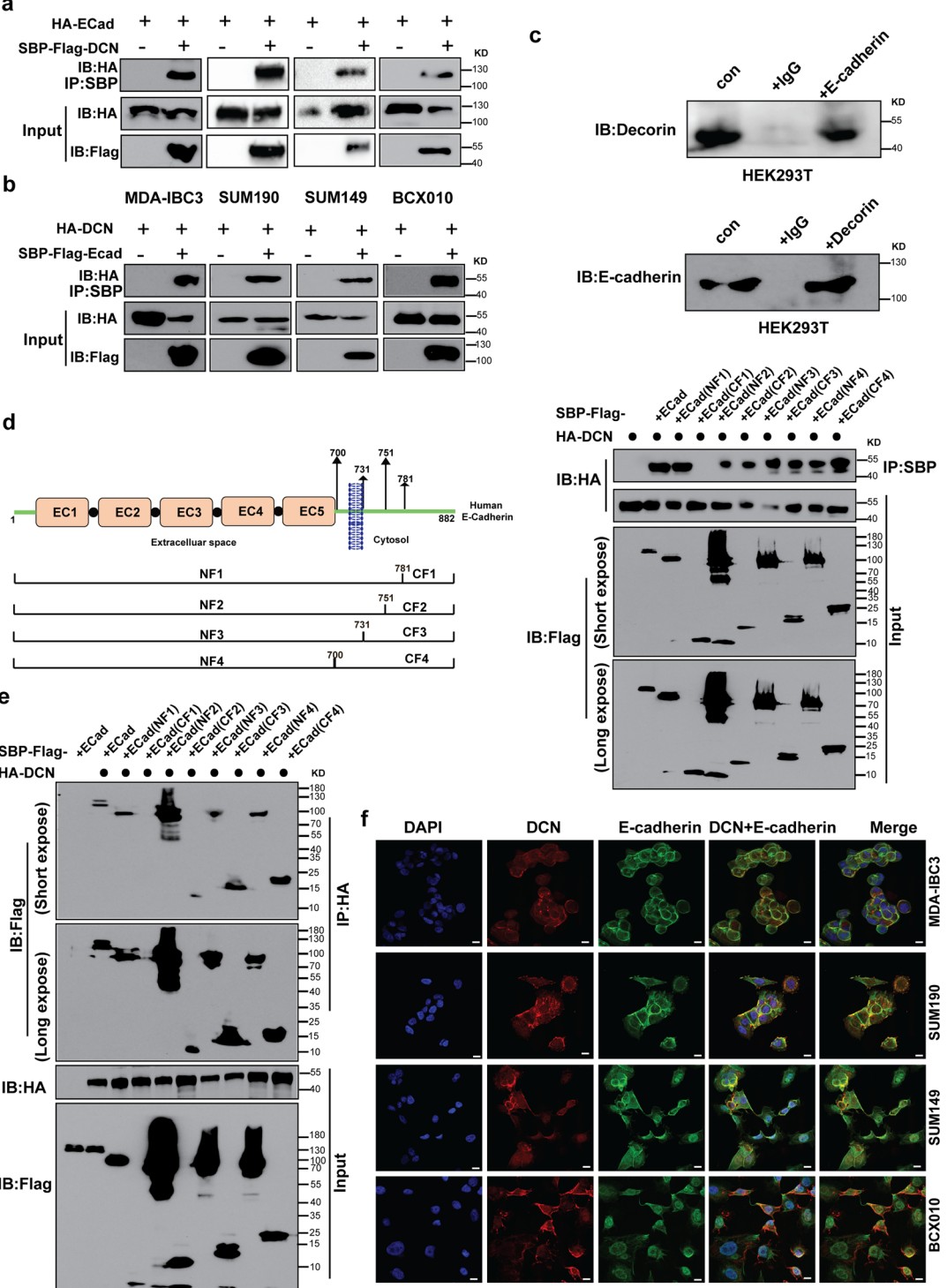

**Fig. 5 DCN interacts physically with E-cadherin in IBC cells. a** and **b** Reciprocal immunoprecipitation assay shows interaction between DCN and E-cadherin in IBC cells. The indicated IBC cell lines were transfected with plasmids expressing HA-tagged DCN or Flag-tagged E-cadherin as indicated. Whole-cell lysates were used for immunoprecipitation with anti-HA antibody or anti-Flag antibody. **c** Endogenous immunoprecipitation assay confirms the interaction between DCN and E-cadherin in HEK293T cells. Lysates of HEK293T cells were subjected to immunoprecipitation with anti-E-cadherin or anti-DCN antibodies. **d** Investigation of the E-cadherin domain required for DCN interaction. A plasmid that expressed SBP-Flag-E-cadherin and its deletion derivatives as indicated in the schematic diagram (left panel) were transfected together with HA-DCN plasmid into HEK293T cells. **d** right and **e** show results of onco-immunoprecipitation indicating a physical interaction between the NF1 domain of E-Cadherin and DCN. **f** Dual immunofluorescence staining visualizes the subcellular localization of E-cadherin (green) and DCN (red) proteins in IBC cell lines under confocal fluorescence microscopy. A merged image of the green/red fluorescence shows co-localization of the proteins on the membrane. Nuclear DNA was visualized by DAPI staining. Scale bar: 30 μm.

A-G agarose (Fig. 5e). Our results revealed that full-length DCN interacted strongly with the NF1 domain of E-cadherin, but did not interact with the CF1 domain. Immunofluorescence confocal microscopy further confirmed that DCN and E-cadherin interacted with each other and co-localized primarily on the membrane of IBC cells (Fig. 5f and Supplementary Figs. 10 and 11). We further confirmed our immunofluorescence observations by subcellular fractionation and immunoblotting. We demonstrated that DCN protein was detected in the cytoplasmic and membrane fractions of IBC cell lines (Supplementary Fig. 12).

**DCN accelerates the degradation of E-cadherin through an autophagy-linked lysosomal pathway.** Next, we investigated the mechanism by which DCN regulates E-cadherin expression in IBC cells. First, we examined if DCN affects E-cadherin mRNA levels. Overexpression of DCN did not affect endogenous E-cadherin mRNA expression in IBC cells (Supplementary Fig. 13). Because DCN affected E-cadherin protein but did not alter E-cadherin mRNA levels, we hypothesized that DCN influences E-cadherin protein expression by regulating its stability. To test this, we treated IBC cells stably transfected with DCN-overexpressing or control vectors with the protein synthesis inhibitor cycloheximide and determined the half-life of E-cadherin protein. We found that DCN overexpression in IBC cells accelerated the degradation of E-cadherin in the presence of cycloheximide (Fig. 6a). Most protein degradation in cells occurs either by lysosomal[45] or proteasome-based proteolytic pathways[46]. To further uncover the mechanisms of the DCN-mediated degradation of E-cadherin protein, we treated DCN-overexpressing IBC cells with the lysosome-function inhibitor chloroquine or the proteasome inhibitor MG132. Inhibition of lysosome function by chloroquine markedly delayed the degradation of E-cadherin that was caused by DCN overexpression in IBC cells (Fig. 6b), but MG132 did not affect the degradation of E-cadherin (Supplementary Fig. 14), indicating that DCN-mediated degradation of E-cadherin protein occurred via the lysosomal pathway. We further investigated which lysosomal pathway (endocytic or autophagic) was involved in DCN-mediated degradation of E-cadherin. Others have reported that E-cadherin can be degraded via the autophagy pathway[47], and DCN has been shown to promote autophagy in endothelial cells by inducing the expression of Beclin1, a well-known autophagy marker and regulator[48,49]. We therefore speculated that DCN promotes E-cadherin autophagic degradation by inducing Beclin1 expression. First, we compared the expression levels of Beclin1 in DCN-overexpressing and control IBC cell lines and found that Beclin1 was significantly upregulated in the DCN-overexpressing IBC cells (Fig. 6c). Next, we investigated whether DCN regulates the stability of E-cadherin through Beclin1-linked autophagy pathway. To test this possibility, we depleted Beclin1 in DCN-overexpressing IBC cells by using two independent shRNAs and analyzed the expression of E-cadherin and EGFR signaling pathway proteins. We found that silencing Beclin1 not only markedly restored the protein level of E-cadherin but also activated the EGFR pathway, as indicated by elevated p-ERK and p-EGFR levels (Fig. 6d). Taken together, our findings demonstrate that DCN acts as a tumor- and metastasis suppressor in IBC by accelerating the autophagic degradation of E-cadherin and suppressing activation of EGFR–ERK signaling (Fig. 6e).

## Discussion

We report here that the proteoglycan protein DCN negatively influences the malignant and aggressive properties of IBC cells. We further showed that DCN reduces the expression of E-cadherin and EGFR, thereby inhibiting activation of EGFR

signaling. We further report the novel finding that DCN interacts with and negatively modulates the stability of E-cadherin in IBC cells via a mechanism involving autophagy-linked lysosomal degradation of E-cadherin.

In the current paradigm, E-cadherin is depicted as a tumor-suppressor gene in a variety of cancers, and its loss is associated with increased invasion and metastatic potential[14,15]. However, other evidence suggests that E-cadherin promotes tumorigenesis in ovarian carcinoma[50], aggressive brain tumors[51], and in breast cancer xenografts generated from MDA-MB-468 cells[52]. Most recently, Ewald and colleagues used several transgenic mouse models of invasive breast cancer to show that E-cadherin is required for the systemic dissemination and metastatic seeding of breast cancer cells to the lung[53]. In IBC, E-cadherin is persistently present in tumor cells, tumor emboli, and metastases[17,20,21]. Functional in vitro experiments showed that a dominant-negative E-cadherin mutant led to decreased invasion and downregulation of EGFR/MAPK signaling in SUM149 IBC cells[20]. In another study, blockade of E-cadherin with antibodies resulted in the loss of homotypic aggregation in Mary-X spheroids in vitro and induced dissolution of metastatic lesions in vivo[17]. Furthermore, the destabilization of E-cadherin through blockade of p120-catenin, which anchors E-cadherin to the cell surface, or silencing of eIF4GI, which regulates translation of specific mRNAs such as p120, reduced tumor growth and formation of tumor emboli in SUM149 cells[11]. In the current study, we demonstrated that silencing E-cadherin or degrading it via DCN overexpression or treatment with recombinant DCN effectively inhibited the robust invasion and aggressiveness exhibited by IBC cells, further cementing E-cadherin as an oncogenic driver and crucial therapeutic target in IBC. Notably, the inhibition of E-cadherin expression by DCN did not induce mesenchymal gene expression in IBC cells, corroborating previous findings that EMT is not the primary means of tumor cell invasion and metastasis in IBC[54,55]. These findings are also consistent with clinical data showing that E-cadherin-negative invasive lobular carcinoma cells remain intrinsically epithelial with limited evidence of EMT features[56] and that E-cadherin loss was insufficient to induce classical EMT in a murine invasive lobular carcinoma model established by conditional mutation of CDH1 and knockout of TP53[57]. Whereas our current findings indicate that DCN is a potent oncosuppressive molecule in IBC cells, as in other solid cancers, we have discovered a novel molecular mechanism that offers new insights into the unique pathobiology of E-cadherin-positive IBC tumors. We discovered that DCN suppresses tumorigenesis and invasion by physically binding with and destabilizing the E-cadherin protein and inhibiting the E-cadherin–EGFR–ERK axis. Previously, DCN was shown to mediate the inhibition of colorectal cancer growth by stabilizing E-cadherin[43], although the mechanisms were not fully understood. To our knowledge, this is the first report to describe a mechanism by which DCN negatively regulates E-cadherin and the E-cadherin–EGFR pathway to attenuate tumorigenesis, invasion, and metastasis. We further showed that DCN mediates attenuation of EGFR signaling in IBC cells even in the presence of EGF ligand stimulation. Further analysis of the ratio of p-EGFR/total EGFR and p-ERK/t-ERK in DCN-overexpressing and control IBC cell lines indicated that the inhibitory effect of DCN on the EGFR–ERK pathway under EGF treatment was mainly dependent on EGFR downregulation, which led to attenuation of its signaling pathway. These observations are supported by earlier studies from Iozzo's group indicating that DCN influences the survival and growth of cancer cells by evoking protracted internalization of EGFR, which leads to EGFR degradation and consequent attenuation of the EGFR signaling cascade[31,58]. Previous studies have shown that DCN is expressed by cancer-associated stromal cells or by cancer

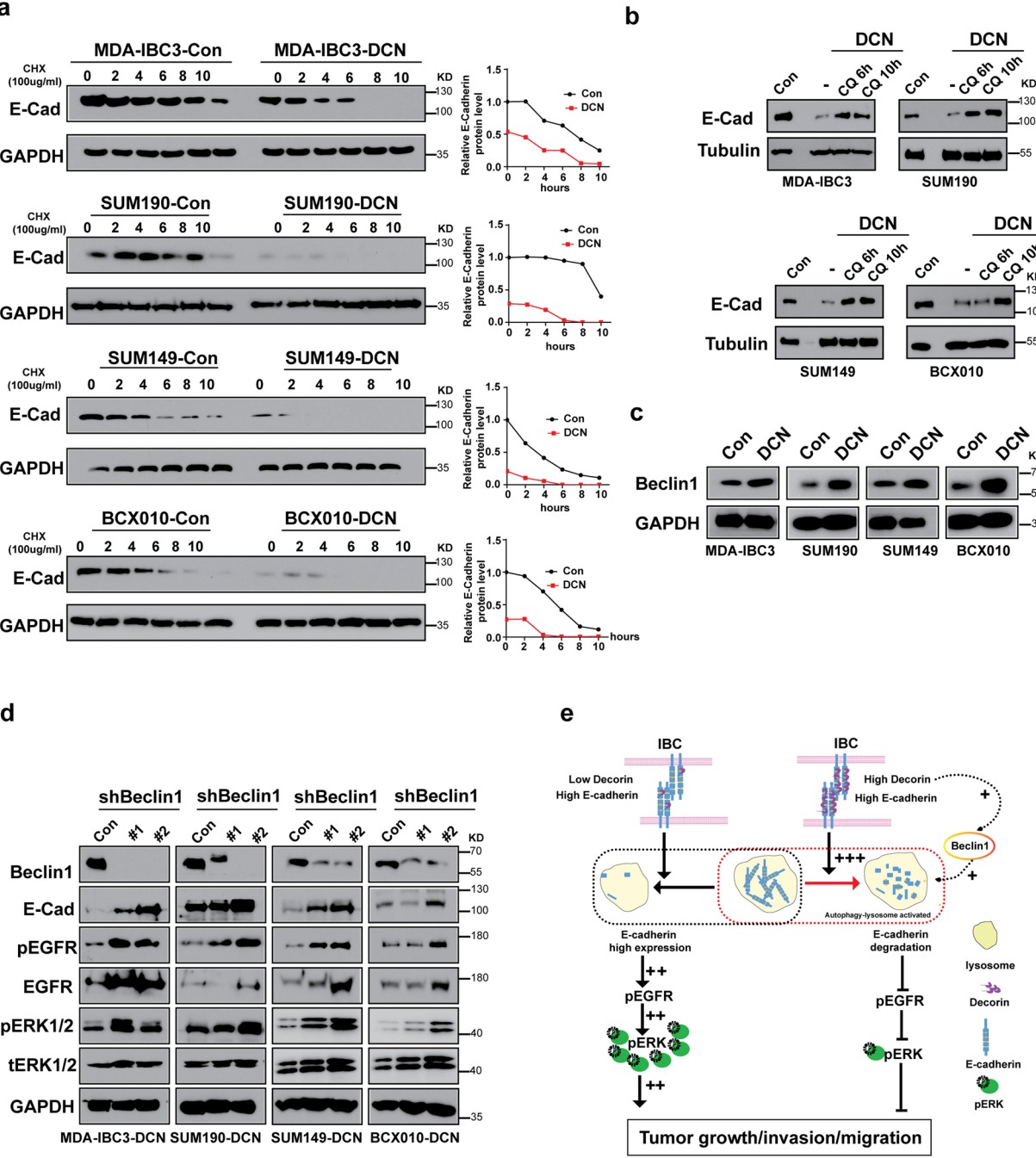

**Fig. 6 DCN modulates the stability of E-cadherin protein through an autophagy-linked lysosomal pathway. a** DCN destabilizes E-cadherin protein in IBC cells. DCN-overexpressing and control IBC cells were treated with the protein synthesis inhibitor cycloheximide (CHX, 100 μg/mL) at 0, 2, 4, 6, 8, or 10 h was shown. Total cell lysates were subjected to western blotting analysis with the indicated antibodies. GAPDH served as a loading control. **b** the lysosomal inhibitor chloroquine (CQ) markedly delayed the DCN-induced degradation of E-Cadherin. DCN-overexpressing IBC cells were treated with 20 μM CQ for 6 or 10 h, and total cell lysates were subjected to western blotting analysis with the indicated antibodies. GAPDH served as a loading control. **c** Overexpression of DCN promotes Beclin1expression in IBC cells. Total cell lysates were subjected to immunoblotting with the indicated antibody. GAPDH served as a loading control. **d** DCN inhibition of E-cadherin expression and EGFR pathway activation in IBC cells is dependent on Beclin1. DCN-overexpressing cells were transduced with two independent lentiviral Beclin1 shRNAs or control shRNA; 48 h later, the cells were harvested and analyzed by western blotting with the indicated antibodies. GAPDH was used as a loading control. Beclin1 knockdown resulted in elevation of E-cadherin expression and EGFR pathway activation in DCN-overexpressing IBC cells. **e** Proposed model summarizing the DCN-mediated inhibition of tumorigenesis and metastasis in IBC cells through a mechanism involving the lysosomal degradation of E-cadherin and suppression of the EGFR–ERK signaling pathway.

cells[59–61]. Analysis of an IBC patient dataset that contains both IBC tumor cells and tumor stroma (GSE5847) confirms that DCN is expressed in both tumor cells and stroma, with slightly higher expression seen in tumor cells. In the current study, we have shown that DCN expression is very low in IBC tumors and

cell lines and that it is localized in the membrane and cytoplasm. As such, the interaction between endogenous DCN and E-cadherin (and/or EGFR) in IBC tumors may not significantly influence ERK signaling activity and IBC cell invasion. However, when ectopically expressed in IBC cells, DCN overexpression or

DCN protein treatment significantly affects the expression of E-cadherin and EGFR–ERK activity as well as the invasion and development of IBC tumors. We cannot exclude the possibility that stroma-derived DCN could also contribute to tumor suppressor function in IBC, because others have linked reduced levels of stromal DCN with more aggressive disease[59,60]. Further characterization of the DCN-expressing tumor cells or stroma cell types, and cell-type-specific knockout of stromal DCN, may be crucial for fully demonstrating the roles of tumor cell or stroma-derived DCN in IBC tumor biology.

Mounting evidence suggests that autophagy has several roles in cancer, promoting tumor invasion by activating EMT and tumor survival by enabling cancer cells to overcome high-energy demands[62,63]. In addition, autophagy can inhibit tumor initiation through clearance of misfolded proteins, reactive oxygen species, and other factors that contribute to genomic instability[64]. DCN has been described as an autophagy-inducible proteoglycan[27]. Indeed, Buraschi et al.[48] demonstrated that DCN induced a pro-autophagic complex consisting of Beclin1 and LC3 in endothelial cells that contributed to inhibition of tumorigenesis and angiogenesis. In addition, Goyal et al. showed that DCN inhibited anti-autophagic signaling via suppression mediated by the Akt/mTOR/p70S6K signaling axis and concurrent activation of the pro-autophagic AMPK-mediated signaling cascade[26]. Our current observations showed that DCN induces the expression Beclin1, a key autophagy marker and regulator, which was also noted by Iozzo's group[48,49]. Moreover, in our study silencing Beclin1 in DCN-overexpressing IBC cell lines led to increased E-cadherin expression and EGFR pathway activation. Hence, we speculate that autophagy acts as a tumor suppressor in IBC cells. Although the role of autophagy in breast cancer tumorigenesis has been described by other groups[65,66], the contributions of Beclin1 and the autophagy pathway in IBC tumor progression and metastasis is unknown. Further investigation is needed to clarify details of the function and molecular mechanism of autophagy in IBC to perhaps reveal important therapeutic targets that can be exploited to combat IBC invasion and tumorigenesis.

Our work provides several novel findings. First, we demonstrated that, in IBC, the inhibitory effect of DCN on the EGFR pathway depends on E-cadherin expression. Second, we discovered that DCN is a novel negative regulator of E-cadherin expression in IBC. Finally, we revealed that DCN activates the autophagy-linked lysosome pathway to destabilize the E-cadherin protein and influence EGFR signaling. Confirming prior findings in other cancer models, DCN also suppresses tumor growth and invasion in IBC models by inhibiting the EGFR/ERK pathway[25,58], and that it interacts physically with E-cadherin as previously shown in a colorectal cell line[43].

In summary, our study provides strong evidence that DCN suppresses aggressive growth, tumorigenesis, and metastasis in IBC via a newly discovered mechanism that involves the degradation of E-cadherin protein and inhibition of the E-cadherin–EGFR axis. Our findings provide new insights and a novel molecular mechanism into the unique biology of IBC tumors and offer promising targets for therapeutic interventions for this aggressive tumor type.

## Methods

**Plasmids**. *Decorin* (Gene ID: 1634) constructed into a modified pLOC-Turbo-RFP plasmid was purchased from the Functional Genomics Core Facility at MD Anderson Cancer Center (Cancer Center Support [Core] Grant NCI P30-CA016672). cDNAs for *Decorin* and *E-cadherin* (Gene ID: 999) were cloned into modified pHA-N1 (#86049, Addgene) and pcDNA3-neo-Strep-flag-Cterm (#102645, Addgene) vectors to generate constructs encoding (hemagglutinin [HA])-tagged DCN and E-Cadherin, and Flag/Strep double-tagged DCN and E-Cadherin. All mutant truncates of E-cadherin were constructed into the pcDNA3-neo-Strep-flag-Cterm plasmid. The pLKO.1-shBeclin1 and control plasmids (TRCN0000299864, TRCN0000299790, and SHC016-Control) were purchased from Sigma (St. Louis, MO, USA). The pLKO.1-shE-cadherin plasmids were obtained from the Functional Genomics Core at MD Anderson. Details of plasmids are given in Supplementary Data 2.

**Antibodies and reagents**. Anti-GAPDH (#8884), anti-tubulin (#2148S), anti-Flag (#8146S), anti-HA-tag (#C29F4) (3724S), anti-Histone H3 (D1H2) (#4499S), anti-Vinculin (#4650), anti-Slug (C19G7) (#9585), anti-Zeb1(#3396), anti-EGFR (D38B1) (#4267), phospho-EGFR (Tyr1068) (#3777S), phospho-p44/42 MAPK (Erk1/2) (Thr202/Tyr204) (#4370S), and p44/42 MAPK (tErk1/2) (#9102) antibodies were purchased from Cell Signaling Technology (Beverly, MA, USA). All secondary antibodies including anti-mouse-horseradish peroxidase (HRP) (#7076S), anti-rabbit-HRP (#7074S), anti-mouse-Alexa Fluor 555 (#4409S), anti-mouse-Alexa Fluor 488 (#4408S), anti-rabbit-Alexa Fluor 594 (#8889S), and anti-rabbit-Alexa Fluor 488 (#4412S) were also purchased from Cell Signaling Technology. Anti-E-cadherin (#MAB1838-SP), anti-N-cadherin (#MAB13881), anti-Fibronectin (#MAB1918), and anti-vimentin (#5741S) antibodies were from R&D Systems (Minneapolis, MN, USA). Anti-Twist (#sc-81417) was purchased from Santa Cruz Biotechnology (Dallas, TX, USA). Anti-Snail (ab53519) was purchased from Abcam (Cambridge, MA, USA). High-capacity streptavidin agarose (#20359), EGF (#phg0314), S-protein agarose (#69704), and protein A/G agarose (#20422) were purchased from Thermo Scientific (Rockford, IL, USA). Recombinant human decorin protein (#143-DE-100) was purchased from R&D Systems (USA). Anti-decorin (#HPA064736), chloroquine (#C6628), and cycloheximide (#C4859) were purchased from Sigma-Aldrich (St Louis, MO, USA). MG-132 (#S8410) was purchased from Selleck (Houston, TX, USA). The prestained Protein ladder (#26616) was purchased form Thermo Scientific (Waltham, MA, USA). Details of antibody dilution are described in the Supplementary Data 2.

**Cell culture transfection**. The SUM149 and SUM190 cell lines were obtained from Asterand (Detroit, MI). MDA-IBC3 cells were generated in our lab[39]. The human TNBC cell line BCX010, derived from a patient with triple-negative IBC, was generously donated by Dr. Funda Meric-Bernstam (MD Anderson Cancer Center)[40]. All IBC cell lines were cultured in Ham's F-12 media supplemented with 10% fetal bovine serum (FBS) (GIBICO Inc., USA), 1 μg/mL hydrocortisone (#H0888, Sigma-Aldrich), 5 μg/ml insulin (#12585014, Thermo Fisher Scientific), and 1% antibiotic–antimycotic (#15240062, Thermo Fisher Scientific). HEK293T cells were obtained from the American Type Culture Collection (Manassas, VA, USA) and were cultured in Dulbecco's modified Eagle's medium (DMEM) supplemented with 10% FBS and 1% penicillin and streptomycin (#15140122, Invitrogen, Carlsbad, CA, USA) at 37 °C in a humidified incubator with 5% $CO_2$. Cell lines were authenticated by STR profiling at the Cytogenetics and Cell Authentication Core at MD Anderson. All the cell lines used in experiments tested negative for mycoplasma contamination using MycoAlert Mycoplasma Detection Kit (#LT07-218, Lonza, ME, USA).

For transfection, cells at 70% confluence were transfected with plasmids by using Lipofectamine3000 (Invitrogen) in serum-free medium according to the manufacturer's instructions. After 6 h of incubation, medium was replaced with fresh complete medium. At 48 h after transfection, cells were collected and subjected to western blotting.

**Lentiviral production and transduction**. To produce high-titer lentivirus, we used standard protocols. Briefly, about $1.2 \times 10^7$ HEK293T cells were plated in 15-cm cell culture dishes in 25 mL Dulbecco's MEM supplemented with 10% FBS. The next day, cells were transfected with Lipofectamine3000 (Invitrogen) DNA mixture (10 μg of pLKO.1 shRNA plasmid/10 μg of pLOC-DCN, 7.5 μg of psPAX2 packaging plasmid and 2.5 μg of pMD2.G envelope plasmid) and were incubated overnight. The culture medium was then removed and replaced with fresh medium. The supernatant containing the virus was then collected, filtered through a 0.45 μm HV Durapore membrane (EMD Millipore, Burlington, MA, USA) to remove cells and large debris, and concentrated by ultracentrifugation.

For transduction, target cells were about 70% confluent. Two hours before transduction, the medium was changed, and then transductions were carried out for 24 h in the presence of 8 μg/mL polybrene (Sigma-Aldrich). Cells expressing fluorescent protein (GFP/RFP) were sorted by fluorescence-activated cell sorting and expanded before in vitro and in vivo experiments.

**Western blotting**. Protein concentrations were measured by using a Pierce BCA Protein Assay Kit (Thermo Scientific). Most experiments were done with 40 μg protein samples, although we loaded 60 μg of protein for studies related to EGFR in SUM190 cells and 60 μg protein to study E-cadherin in BCX010 cells. Proteins were separated by sodium dodecyl sulfate polyacrylamide gel electrophoresis (SDS–PAGE) and transferred onto polyvinylidene fluoride membrane by using a semi-dry transfer unit (both from Bio-Rad, Hercules, CA, USA). After being blocked in Tris-buffered saline containing 5% non-fat milk and 0.1% Tween-20, the membranes were incubated with primary antibodies for 2 h at room temperature, and subsequently probed with HRP-conjugated secondary antibodies at room

temperature for 1 h. Immunoreactivity was visualized with the ECL chemiluminescence system (Bio-Rad).

**Quantitative real-time PCR**. Total RNA was extracted from tissue or cell culture samples by using the Trizol reagent (#A33251, Invitrogen) according to the manufacturer's instructions, and 1–2 μg of RNA was treated with RNase-free DNase (#AM1906, Invitrogen) to remove genomic DNA contamination. Oligonucleotide primer sequences used in the reverse transcriptase-PCR are listed in Supplementary Data 2. qRT-PCR analysis was conducted with a SYBR Green Supermix kit (#4368706) with ABI7500 (both from Applied Biosystems, Foster City, CA, USA). The cycle parameters were 95 °C for a 1 min hot start and 45 cycles of 95 °C for 10 s, 60 °C for 10 s, and 72 °C for 20 s. The fold change in expression was calculated using the $2^{-\Delta\Delta Ct}$ method, with glyceraldehyde-3-phosphate dehydrogenase (GAPDH) mRNA as an internal control. Experiments for each sample were done in triplicate.

**Cell proliferation**. To assess cell proliferation, 1000 cells were seeded in 96-well plates with five replicates and the cell growth capacity was measured every day with the CellTiterBlue assay (#G8080, Promega, Madison, WI, USA) according to the manufacturer's instructions. Absorbance was recorded at OD560/OD590 nm with a Multifunctional Reader VICTOR X 3 (PerkinElmer, Waltham, MA, USA).

**Colony formation assay**. To assess colony formation, 2000 cells (MDA-IBC3, SUM190) or 500 cells (SUM149, BCX010) were seeded in six-well plates in triplicate. Medium was replenished every 3 days. For the DCN protein treatment, 500 cells (MDA-IBC3) or 200 cells (SUM149, SUM190, BCX010) were seeded in 24-well plates in triplicate. Medium was replenished every 3 days including the DCN (8 μg/mL). After 14 days, colonies were fixed with 4% paraformaldehyde (#AAJ19943K2, Thermo Scientific) and stained with 1.0% crystal violet. The numbers of colonies were counted with Image J software (National Institutes of Health, Bethesda, MD, USA).

**In vitro migration and invasion assays**. Cell migration was measured in 24-well transwell plates (Corning, Inc., Corning, NY, USA) as follows. Cells ($5 \times 10^4$ in 100 μL serum-free medium) were seeded into the upper chamber, and 800 μL of serum-containing medium was used in the lower chamber as the attractant. After 24 h of culture, cells that had migrated to the bottom surface were fixed in 4% paraformaldehyde (in phosphate-buffered saline [PBS]) for 30 min and stained with 1% crystal violet solution for 25 min, and the non-migrated cells were gently removed from the upper chamber with a cotton swab. Under microscopy, 10 randomly chosen visual fields were recorded and analyzed with ImageJ software. For invasion assays, the upper chamber was pre-coated with Matrigel (BD Biosciences, Franklin Lakes, NJ, USA), and $5 \times 10^4$ cells were seeded into the upper chamber.

**Mammosphere assay**. Primary mammospheres were formed by culturing $4–5 \times 10^4$ cells/well in ultra-low-attachment six-well plates (Corning, Inc.) in MEM that contained aliquots of recombinant epidermal growth factor (rEGF), fibroblast growth factor [FGF]-Basic, B27, and Gentamycin/Pen-strep B (all from Thermo Fisher, Waltham, MA, USA). For the secondary mammosphere formation assay, cells from monolayer culture were cultured as primary mammospheres in a 10-cm ultra-low-attachment dish (Corning, Inc.) for 4 days, and then those mammospheres were collected and dispersed with 0.05% trypsin, seeded in six-well ultra-low-attachment plates (10,000 cells/well) in mammosphere media, and counted a week later after having been stained with MTT (0.4 mg/mL, Sigma-Aldrich) to improve visualization of spheres. The number of spheres exceeding 80 μm in diameter was counted with a GelCount device (Oxford Optronix, Oxford, UK). Spheroid-forming efficiency (SFE) was calculated as the total number of secondary mammospheres (with a diameter larger than 80 μm) divided by total number of cells plated in serial dilutions of cell numbers. To evaluate the spheroid average size, 100 cells were plated in each well, and images of spheroid size were obtained with a Nikon eclipse Ti camera (NY, USA).

**Analysis of gene expression**. The online database Gene Expression across Normal and Tumor Tissues (GENT), which includes more than 34,000 samples, was used to compare DCN expression in normal and cancer tissues across tumor types. Expression of DCN in normal breast versus breast tumors versus tumor stroma was assessed with The Cancer Genome Atlas (TCGA) Breast Cancer Project, Richardson (GSE3744), Hatzis (GSE25066), Sircoulomb (GSE17907), and Boersma (GSE5847) datasets. To generate Kaplan–Meier curves, we used Gene Expression Based Outcome (GOBO) and IBC Consortium datasets. For each dataset, patients were stratified as DCN High or DCN Low according to the median DCN expression in the tumor samples within that dataset.

**Xenograft studies**. Animal experiments were done in accordance with protocols approved by the Institutional Animal Care and Use Committee of MD Anderson Cancer Center, and mice were killed when they met the institutional euthanasia criteria for tumor size and overall health condition. Four- to six-week-old female athymic SCID/Beige mice were purchased from Harlan Laboratories (Indianapolis,

IN, USA). Animal care and use were in accordance with institutional and NIH guidelines. Xenografts were created by injecting $5 \times 10^5$ SUM149-Con/DCN or MDA-IBC3-Con/DCN cells (9 mice/group) into the cleared mammary fat pad of the SCID/Beige mice[42], and tumor growth was monitored with calipers. Tumor volumes were assessed weekly (MDA-IBC3) or twice weekly (SUM149) by measuring the externally apparent tumors in two dimensions with calipers. Volume ($V$) was determined by the following equation, where $L$ is the length and $W$ is the width of the tumor: $V = (L \times W^2) \times 0.5$. For in vivo metastasis studies, $1 \times 10^6$ SUM149-Control or DCN-overexpressing cells were injected into the lateral tail veins of 4–6-week-old female SCID/Beige mice ($n = 10$ per group).

**Immunohistochemistry staining**. Formalin-fixed, paraffin-embedded sections of primary tumors were stained with hematoxylin and eosin and used for immuno-histochemical staining to detect E-cadherin, DCN, and EGFR. All staining was done by the flow cytometry and Cellular Imaging Core facility at MD Anderson with standard, validated protocols, and the findings were analyzed by a pathologist specializing in breast cancer (S.K.). Images were acquired with a LEICA DC500 camera (Solms, Germany).

**Immunoprecipitation**. Immunoprecipitation experiments were done as follows[67]. HEK293T and IBC cells were seeded 24 h before transfection. Cells were transiently transfected by Lipofectamine3000 (Invitrogen), collected 24 h later, and then lysed in 1× RIPA buffer (Sigma-Aldrich) containing 10 μL/mL phosphatase inhibitor cocktail and 10 μL/mL protease inhibitor cocktail (Santa Cruz Biotechnology) for 10 min. For exogenous immunoprecipitation with streptavidin beads, suspensions of streptavidin beads and cell lysates were incubated for 2.5 h on an orbital shaker at 4 °C. After five washes with 1× RIPA buffer, the pellets were resuspended in SDS sample buffer and boiled for 8 min.

For endogenous immunoprecipitation, HEK293T cell lysates were incubated with 25 μL protein A/G beads and anti-E-cadherin or anti-DCN antibody for 4 h on an orbital shaker at 4 °C. After centrifugation, the pellets were washed five times in 1× RIPA buffer, resuspended in 2× SDS sample buffer, and boiled for 8 min[67].

**ERK activation analysis**. To analyze ERK activation, cells were starved in serum-free medium for 36 h, after which EGF was added to the medium to a final concentration of 50 ng/mL to stimulate activation of the EGFR/MAPK pathway. Cells were then harvested at 0, 5, 10, 20, 30, and 60 min.

**Immunofluorescence**. Cells were grown on Millicell EZ SLIDE four-well glass (PEZGS0416) and transiently transfected with 1.5 μg of the indicated expressing plasmids. After 24 h, cells were fixed in 4% paraformaldehyde (in PBS) for 15 min at room temperature. After three washes in 1× PBS, the coverslips were incubated in 0.5% Triton X-100 (in PBS) for 15 min at room temperature, rinsed again in PBS, and blocked in 1% goat serum (in PBS) for 1 h at room temperature. HA-DCN was stained with anti-HA antibody (1:200 dilution) for 2 h at room temperature and visualized with an Alexa-Fluor 555-conjugated secondary antibody (1:500 dilution). Flag-E-cadherin was stained with anti-Flag antibody (1:500 dilution) for 2 h at room temperature and visualized with an Alexa-Fluor 468-conjugated secondary antibody (1:500 dilution). Nuclei were labeled with 4′6-diamino-2-phenylindole (DAPI) for 15 min at room temperature. Antibodies against DCN (Sigma-Aldrich) and E-cadherin (R&D) were used for the endogenous immunofluorescence. Finally, the coverslips were washed five times in 1× PBS. Immunofluorescence microscopy images were obtained a Zeiss LSM710 laser confocal microscope (Carl Zeiss Micro Imaging, Inc., Jena, Germany).

**Subcellular protein fractionation**. To study the localization of DCN in cultured IBC cells we used the Subcellular Protein Fractionation Kit (#78840, Thermo Scientific, USA) to isolate the cytoplasmic (CEB), membrane (MEB), and nuclear (NEB) fractions. The samples were processed, and proteins were detected by western blotting. Vinculin, EGFR, and Histone H3 served as loading control markers for the cytoplasmic, membrane, and nuclear fractions, respectively.

**Statistics and reproducibility**. All experiments were repeated at least three times. Unless otherwise noted, all graphs depict mean ± SEM. Statistical significance was determined with Student's $t$ tests (unpaired, two-tailed) unless otherwise specified (Fisher's exact test or $X^2$ text). Biostatistical analyses were done with Office SPSS software (SPSS version 22.0, IBM, New York, NY, USA) and GraphPad software (GraphPad Prism 8, La Jolla, CA, USA).

**Reporting summary**. Further information on research design is available in the Nature Research Reporting Summary linked to this article.

## Data availability
All data are available in the main and supplementary files or available from corresponding author upon reasonable request. Uncropped blots of main figures and Supplementary figures are shown in Supplementary Fig. 15. All other data supporting the findings of the study are available within the paper and Supplementary information.

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

## Acknowledgements

We thank Jeff Rosen (Baylor College of Medicine) for critical review of the manuscript and helpful discussions and Christine F. Wogan, MS, ELS, of MD Anderson's Division of Radiation Oncology for scientific review and editing of the manuscript. We thank Carol M. Johnston from the Division of Surgery Histology Core at MD Anderson for immunohistochemical analysis. We thank the Flow Cytometry and Cellular Imaging Core Facility at MD Anderson Cancer Center for confocal microscopy and technical support with flow sorting. This work has been supported by funding from the Susan G. Komen Career Catalyst Research grant (CCR16377813), an American Cancer Society Research Scholar grant (RSG-19-126-01) to B.G.D., MD Anderson Cancer Center (startup fund), the State of Texas Rare and Aggressive Breast Cancer Research Program, and Cancer Center Support (Core) Grant P30 CA016672 from the National Cancer Institute, National Institutes of Health, to The University of Texas MD Anderson Cancer Center.

## Author contributions

X.H. and B.G.D. conceived and designed the project. X.H. performed most of the experiments, analyzed the data, and interpreted the results. E.S.V., O.M.R., and R.L. performed some experiments. Y.G. provided and analyzed tissue microarrays of IBC human patient samples. J.S. provided statistical analysis support. S.K. provided pathological expertise and analysis of xenograft tumors. X.W., N.T.U., D.T., and W.A.W. provided resources and contributed to revision of the manuscript. X.H. and B.G.D. wrote and edited the manuscript with input from all other authors. B.G.D. supervised the study.

## Competing interests

The authors declare no competing interests.
