## [Peer Review File · Communications Biology]

REVIEWERS' COMMENTS:

Reviewer #1 (Remarks to the Author):

Summary: Decorin is a leucine-rich proteoglycan produced by a variety of stromal cells in the body and also in some breast cancer-associated stromal cells. Many studies of tumor-suppressor function of DCN has been largely attributed to its actions as a negative regulatory ligand for multiple tyrosine-kinase receptors, in particular EGFR, also inhibition of angiogenesis by binding VEGF-R2 and thrombospondin. Earlier studies have also shown that expression of E-cadherin, commonly associated with inflammatory breast cancer (IBC), is (paradoxically) a tumor-promoting event, by virtue of stimulating EGFR pathway. This manuscript demonstrates a novel mechanism for tumor-suppressor functions of DCN in inflammatory breast cancer (IBC) by binding to E-cadherin (E-cadh), a cell membrane protein, and promoting its degradation by lysosome-mediated autophagy. The authors also confirm earlier reports that DCN downregulated EGFR signalling and now link it to E-Cadh degradation. However there are several issues, in particular, the cellular location of the DCN protein in IBC, that needs to be clarified. They should address some deficiencies in their data presentation and answer some questions, as outlined below:

(1) The author used four different IBC lines having different phenotypes. They resorted to both ectopic DCN over-expression and exogenous addition of DCN in functional assays. However in some experiments (migration and invasion assays) they used only two lines (Sum149, BCXOIO), (Fig 1, C, D). They mention in the text that the other two lines (MDA-IBC3 and Sum190) were non-migratory and non-invasive in vitro. This is somewhat surprising since both lines were tumorigenic in orthotopic xenografts. This discrepancy suggests that either a small subpopulation (such as EGFR-expressing subset) grew as tumors or some paracrine factors derived from host stromal cells allowed them to grow. The authors should provide some evidence-based reasons or at least conduct migration/invasion assays in the presence of exogenous EGF.

(2) Figs 1e,f, They used mammosphere numbers for comparisons. Apparently the numbers are generated by plating a fixed cell number. They should conduct mammosphere assays with limiting dilutions of cell numbers to verify that each sphere is a clone of tumor stem cells and then should express the values as sphere-forming efficiency (SFE) for comparing results. Since they show that DCN over-expression did not affect cell proliferation (Extended data Fig 3), they should also check whether the rate of mammosphere growth (average sphere-size, -measurable as perimeters or diameters at specific time periods) was reduced or not (see Tordjman J et al BMC Cancer, 19, 2019). From the pictures presented in Fig 1. it appears that the mean sphere size also dropped in DCN-overexpressing cells. Demonstration of a reduction in the rate of sphere growth should allow them to make an important conclusion- that although DCN did not influence the over-all growth of the cell lines, it reduced the self-renewal capacity of the cell lines in vitro.

(3) Fig 1 G-I indicates that exogenous DCN reduced migration/invasion of IBC cell lines. Q. Does exogenous DCN also reduce mammosphere-forming efficiency?

(4) Fig 2. MDA-IBC3 control tumors show weak DCN staining at or near the cell membranes. Is this pericellular staining or staining of DCN in the cell membrane? This issue is important since DCN is typically a product mesenchymal cells, found in the ECM, and is occasionally reported in in tumor cells as

an aberrant phenomenon. (see related comment 8)

(5) Fig 3 a,b The western blot results indicate a reduced p-EGFR as well as total EGFR proteins in DCN overexpressing cells in vitro. The authors should also plot p-EGFR/total EGFR ratios to find out whether EGF-R signaling pathway is compromised by EGFR down-regulation rather than phosphorylation. This is important in view of earlier studies by the Izzo lab showing the EGFR phosphorylation by DCN is transient, following DCN mediated EGFR degradation by the caveolar pathway, thus impeding EGFR recycling to the cell membrane

Similarly, P-ERK/total ERK ratios should be plotted to validate reduced ERK activation. Fig 3 c (in vivo data): better immunoblots are needed for E-cadherin.

Fig 3g: similar to in vitro data, p-EGFR/total EGFR ratios should also be plotted

Fig 3 h. In the present context that E-cadherin (an epithelial marker) is paradoxically upregulated in IBC, the authors should test a few other additional EMT markers such as N-cadherin, Snail, Slug, Zeb1 and Twist proteins to test whether they also remain un-affected in DCN over-expressing cells.

(6) Fig 4. E-cadherin KD cells show reduced p-EGFR, without affecting total EGFR.

This should be better illustrated by plotting pEGFR/total EGFR ratios

Fig 4 e,f The plots should be shown as Sphere forming efficiency (SRE) as pointed out earlier.

(7) In the discussion section, the authors should clarify what is novel and what is confirmation of earlier reports.

(8) The puzzle regarding the location of DCN protein in tumors.

Their cancer data mining of DCN gene expression does not discriminate between tumor cells and tumor stroma. Thus they should clarify whether DCN is found to be a tumor suppressor when highly expressed by breast cancer-associated stromal cells or cancer cells? Examples of expression by stromal cells: Oda, G., T. Sato, T. Ishikawa, H. Kawachi, T. Nakagawa, T. Kuwayama, et al. 2012. Significance of stromal decorin expression during the progression of breast cancer. *Oncol. Rep.* 28:2003–2008. Hong X et al Reduced Decorin Expression in the Tumor Stroma Correlates With Tumor Proliferation and Predicts Poor Prognosis in Patients With I-III A Non-Small Cell Lung Cancer. *Tumor Biology*, 2016). On the other hand, aberrant DCN expression has been reported in the nuclei of dysplastic oral epithelium and oral squamous cell carcinomas (Banerjee A et al. Aberrant expression and localization of decorin in human oral dysplasia and squamous cell carcinoma, *Cancer Res* 63, 7769-7776, 2003). In Supplementary Fig 1g they show immunostaining in a tissue microarray (TMA), but the location of immunostaining is undefined. They should specify the location of DCN staining.

In their immunostaining figures of wild type (DCN-low) IBC cell lines (supplementary Extended Fig 7), they show DCN staining in the cell membrane co-localized with E-cadherin which is a cell membrane protein. It is rather strange that E-cadherin - believed to be a tumor-promoter protein in IBC, is binding to be a tumor-suppressor protein DCN in aggressive IBC cell lines! Conceptually I find it difficult since they had to introduce DCN ectopically to counter their aggressive functions. The authors should clarify whether this a tumor-associated aberrant phenomenon. Since most of their functional data are derived with ectopic DCN over-expressing IBC cell lines, they should further specify where the DCN protein localized in these cells is. In their figures 3 e,f. the staining appears to be cytoplasmic.

They should devote a short section in the discussion to clarify this puzzle of how DCN in the cell membrane in low DCN-expressing IBC lines works as a tumor suppressor by binding to E-cadherin.

Reviewer #2 (Remarks to the Author):

Inflammatory breast cancer (IBC) is one of highly aggressive and metastatic form of breast cancer. It is essential not only to understand the underlying molecular mechanisms contributing to the aggressive nature of the IBC but also to develop novel and more effective therapeutic interventions for this aggressive form of breast cancer. Decorin (DCN) has been shown to be a tumour suppressor in different types of cancer including breast cancer. In this study, Hui and colleagues focused on IBC and report that 1) DCN is downregulated in tumors from patients with IBC, 2) overexpression of DCN in IBC cells is accompanied by decreased migration, invasion, and cancer stem cells in vitro and inhibited IBC tumor growth and metastasis in vivo, and 3) DCN function as a suppressor of tumour invasion and growth in IBC. They also have shown that these effects were mediated by reducing E-cadherin expression and inhibiting EGFR/ERK signalling pathways. Indeed, DCN binding to E-cadherin in IBC cells were found to accelerate its degradation through an autophagy-linked lysosomal pathway. However, DCN-mediated inhibition of E-Cadherin was found to be insufficient for the induction of EMT in IBC cells. On the basis of their data presented, the authors concluded that DCN induced inhibition of tumorigenesis and metastasis in IBC cells is mediated via downregulation of the E-cadherin/EGFR/ERK axis.

Overall, this is an interesting study and the results presented suggest a novel mechanism of IBC pathobiology and potential targets for therapeutic intervention for this type of breast cancer.

Reviewer #3 (Remarks to the Author):

The manuscript written by Xiaoding Hu et al. is a well planned work with interesting findings. The exhibited results are benefit in breast cancer research field. I think it is suitable to be accepted to submit in the journal.

Point-by-point response to editor's and reviewer's comments

Thank you for the opportunity to resubmit our above-referenced paper. We appreciate the reviewers' overall enthusiasm for the novelty and importance of our findings. We agreed with all of the reviewers' comments and constructive suggestions and have addressed each of their comments, performing additional experiments as needed. We believe the paper has been significantly strengthened by this revision. Reviewer comments, with detailed responses to each comment, are provided below.

REVIEWERS' COMMENTS

Reviewer 1 (Remarks to the Author):

Summary: Decorin is a leucine-rich proteoglycan produced by a variety of stromal cells in the body and also in some breast cancer-associated stromal cells. Many studies of tumor-suppressor function of DCN has been largely attributed to its actions as a negative regulatory ligand for multiple tyrosine-kinase receptors, in particular EGFR, also inhibition of angiogenesis by binding VEGF-R2 and thrombospondin. Earlier studies have also shown that expression of E-cadherin, commonly associated with inflammatory breast cancer (IBC), is (paradoxically) a tumor-promoting event, by virtue of stimulating EGFR pathway. This manuscript demonstrates a novel mechanism for tumor-suppressor functions of DCN in inflammatory breast cancer (IBC) by binding to E-cadherin (E-cadh), a cell membrane protein, and promoting its degradation by lysosome-mediated autophagy. The authors also confirm earlier reports that DCN downregulated EGFR signalling and now link it to E-Cadh degradation. However there are several issues, in particular, the cellular location of the DCN protein in IBC that needs to be clarified. They should address some deficiencies in their data presentation and answer some questions, as outlined below:

(1) The author used four different IBC lines having different phenotypes. They resorted to both ectopic DCN over-expression and exogenous addition of DCN in functional assays. However in some experiments (migration and invasion assays) they used only two lines (Sum149, BCXO10), (Fig 1, C, D). They mention in the text that the other two lines (MDA-IBC3 and Sum190) were non-migratory and non-invasive in vitro. This is somewhat surprising since both lines were tumorigenic in orthotopic xenografts. This discrepancy suggests that either a small subpopulation (such as EGFR-expressing subset) grew as tumors or some paracrine factors derived from host stromal cells allowed them to grow. The authors should provide some evidence-based reasons or at least conduct migration/invasion assays in the presence of exogenous EGF.

Response: We agree that the MDA-IBC3 and SUM190 IBC cell lines are tumorigenic in mouse orthotopic xenografts but are non-migratory and non-invasive in vitro, and we agree that this discrepancy could be attributable to the growth features of the cancer cells in vitro and the contribution of the host organ-specific tumor microenvironment in vivo.

Previous studies of breast and colon cancer cell lines revealed that epithelioid cells that are non-invasive in vitro can be invasive and even metastatic in vivo (Noel et al, Cancer Research 1991; de Both et al, BJC 1999); specifically, the in vitro invasive capacity of cancer cells was shown to correlate with

having a spindle cell shape, vimentin expression, and E-cadherin downregulation. These features are consistent with the features observed in the non-invasive MDA-IBC3 and SUM190 cell lines, which grow in vitro as epithelioid cells that express high E-cadherin but little to no vimentin. A transient downregulation of E-cadherin could also occur in vivo for these IBC cancer cell to be invasive.

At this reviewer's suggestion, we conducted migration and invasion assays with MDA-IBC3 and SUM190 cells in the presence of exogenous EGF in vitro. The **figure at right** shows that EGF alone did not improve the migratory or invasive capacity of these cells.

References

Noël AC, A Callé, H P Emonard, B V Nusgens, L Simar, J Foidart, C M Lapiere, J M Foidart. Invasion of reconstituted basement membrane matrix is not correlated to the malignant metastatic cell phenotype. *Cancer Res* 51(1):405-14 (1991).

de Both NJ, Vermey M, Dinjens WN, Bosman FT. A comparative evaluation of various invasion assays testing colon carcinoma cell lines. *Br J Cancer* 81(6):934-41 (1999).

(2) Figs 1e, f, they used mammosphere numbers for comparisons. Apparently, the numbers are generated by plating a fixed cell number.

They should conduct mammosphere assays with limiting dilutions of cell numbers to verify that each sphere is a clone of tumor stem cells and then should express the values as sphere-forming efficiency (SFE) for comparing results. Since they show that DCN over-expression did not affect cell proliferation (Extended data Fig 3), they should also check whether the rate of mammosphere growth (average sphere-size, -measurable as perimeters or diameters at specific time periods) was reduced or not (see Tordjman J et al *BMC Cancer*, 19, 2019). From the pictures presented in Fig 1. It appears that the mean sphere size also dropped in DCN-overexpressing cells. Demonstration of a reduction in the rate of sphere growth should allow them to make an important conclusion- that although DCN did not influence the over-all growth of the cell lines, it reduced the self-renewal capacity of the cell lines in vitro.

Response: Thank you for raising these interesting points. We addressed this issue by conducting additional experiments with the mammosphere assay protocol and plated serial dilutions of cell numbers (10,000, 1000, 100 cells) to compare the sphere-forming efficiency (SFE) and spheroid size between DCN-overexpressing and control groups. We found that the SFE was significantly reduced in DCN-overexpressing IBC cell lines relative to control cell lines (**Figure 1f and Supplementary Fig 4a**). We also plated DCN-expressing cells or control cells (100 per well) in mammosphere medium for 7 days, after which we found that the average diameter of spheres was much smaller in the DCN-overexpressing cell lines (**Supplementary Fig 4b**), indicating that DCN reduced the self-renewal capacity of the IBC cell lines. Notably, the DCN-expressing cells formed spheroids of <80 μm diameter, which was below the detection limit of 80 μm that we set for the Gel Count machine (per the mammosphere protocol).

(3) Fig 1 G-I indicates that exogenous DCN reduced migration/invasion of IBC cell lines. Q. Does exogenous DCN also reduce mammosphere-forming efficiency?

Response: We treated the four parental IBC cell lines with DCN protein to a concentration of 8 $\mu\text{g}/\text{mL}$ (with DCN protein added every 2 days based on the protein's half-life). We found that this treatment with DCN protein significantly inhibited the number of primary mammospheres (**Supplementary Fig 4c**). We further determined the effects of DCN protein on sphere-forming efficiency and spheroid size by plating serial dilutions of IBC cell numbers (1000, 100 cells). The results shown in **Supplementary Fig 4d-**

e demonstrate that treatment with DCN protein significantly reduced both the sphere-forming efficiency and the average size of the spheroids, which is consistent with the findings from DCN-overexpressing IBC cells.

(4) Fig 2. MDA-IBC3 control tumors show weak DCN staining at or near the cell membranes. Is this pericellular staining or staining of DCN in the cell membrane? This issue is important since DCN is typically a product mesenchymal cells, found in the ECM, and is occasionally reported in in tumor cells as an aberrant phenomenon. (See related comment 8)

Response: Thank you for raising this interesting point, which we address in full in our response to comment 8 below.

(5) Fig 3 a,b The western blot results indicate a reduced p-EGFR as well as total EGFR proteins in DCN overexpressing cells in vitro. The authors should also plot p-EGFR/total EGFR ratios to find out whether EGF-R signaling is pathway is compromised by EGFR down-regulation rather than phosphorylation. This is important in view of earlier studies by the Iozzo lab showing the EGFR phosphorylation by DCN is transient, following DCN mediated EGFR degradation by the caveolar pathway, thus impeding EGFR recycling to the cell membrane. Similarly, P-ERK/total ERK ratios should be plotted to validate reduced ERK activation.

Response: Thank you for this great suggestion. We quantified the p-EGFR/total EGFR and p-ERK/t-ERK ratios and provided the findings in **Figures 3a and 3b**. Our results showed that DCN overexpression and treatment with DCN protein inhibited EGFR-ERK signaling in IBC cells, which depended mainly on the reduction of total EGFR expression.

Fig 3 c (in vivo data): better immunoblots are needed for E-cadherin.

Response: The E-cadherin immunoblot has been updated (**updated Figure. 3c**).

Fig 3g: similar to in vitro data, p-EGFR/total EGFR ratios should also be plotted

Response: We also analyzed the ratio of p-EGFR/total EGFR and p-ERK/t-ERK in DCN-overexpressing and control IBC cell lines treated with EGF; updated results are now shown in **Supplementary Fig 6a and b**. We found that the inhibitory effect of DCN on the EGFR-ERK pathway under EGF treatment depended mainly on downregulation of total EGFR expression (especially after 60 minutes of EGF treatment). We have included mention of this finding in the Discussion section of the revised manuscript.

Fig 3 h. In the present context that E-cadherin (an epithelial marker) is paradoxically upregulated in IBC, the authors should test a few other additional EMT markers such as N-cadherin, Snail, Slug, Zeb1 and Twist proteins to test whether they also remain un-affected in DCN over

Response: Thank you for the suggestion. To confirm whether DCN affects protein expression of other EMT family members, we measured levels of fibronectin, N-cadherin, vimentin, Snail, Slug, Zeb1, and Twist in DCN-overexpressing and control stable cell lines. We found that in IBC cells, ectopically expressed DCN significantly downregulated E-cadherin protein levels but did not affect the expression of other EMT markers. The updated results are shown in **Figure 3h**.

(6) Fig 4. E-cadherin KD cells show reduced p-EGFR, without affecting total EGFR. This should be better illustrated by plotting pEGFR/total EGFR ratios

Response: The pEGFR/total EGFR ratios have been quantified and shown in **Figure 4d**.

Fig 4 e,f The plots should be shown as Sphere forming efficiency (SRE) as pointed out earlier.

Response: As suggested by the reviewer, we assessed the sphere-forming efficiency of E-cadherin knockdown and control IBC cells by plating serial dilutions of cell numbers (1000 or 100 cells). We have included new findings showing that depletion of E-cadherin in IBC cells significantly inhibited sphere-forming efficiency and reduced the average sphere size (**Supplementary Fig 8a, 8b**).

(7) In the discussion section, the authors should clarify what is novel and what is confirmation of earlier reports.

Response: We have endeavored to clarify which of our findings are novel and which are confirmatory throughout the revised manuscript.

(8) The puzzle regarding the location of DCN protein in tumors.

Their cancer data mining of DCN gene expression does not discriminate between tumor cells and tumor stroma. Thus they should clarify whether DCN is found to be a tumor suppressor when highly expressed by breast cancer-associated stromal cells or cancer cells? Examples of expression by stromal cells: Oda, G., T. Sato, T. Ishikawa, H. Kawachi, T. Nakagawa, T. Kuwayama, et al. 2012. Significance of stromal decorin expression during the progression of breast cancer. *Oncol. Rep.* 28:2003–2008. Hong X et al Reduced Decorin Expression in the Tumor Stroma Correlates With Tumor Proliferation and Predicts Poor Prognosis in Patients With I-III A Non-Small Cell Lung Cancer. *Tumor Biology*, 2016). On the other hand, aberrant DCN expression has been reported in the nuclei of dysplastic oral epithelium and oral squamous cell carcinomas (Banerjee A et al. Aberrant expression and localization of decorin in human oral dysplasia and squamous cell carcinoma, *Cancer Res* 63, 7769-7776, 2003).

Response: We agree that our data mining of DCN gene expression did not distinguish between tumor cells and tumor stroma; however, we were able to analyze another IBC patient dataset (GSE5847) that contains both IBC tumor cells and tumor stroma. We found that DCN mRNA was expressed in both cancer cells and stroma, with higher expression seen in the tumor cells (**Supplementary Fig 1g**).

In Supplementary Fig 1g they show immunostaining in a tissue microarray (TMA), but the location of immunostaining is undefined. They should specify the location of DCN staining.

Response: We have now provided immunostaining images from IBC tissue microarray, confirming the localization of DCN on the cell membrane and in the cytoplasm of IBC tumors (**updated from Supplementary Fig 1g to Supplementary Fig 1h**). The images and results were confirmed by the pathologist (Y.G.).

In their immunostaining figures of wild type (DCN-low) IBC cell lines (supplementary Extended Fig 7), they show DCN staining in the cell membrane co-localized with E-cadherin which is a cell membrane protein. It is rather strange that E-cadherin - believed to be a tumor-promoter protein in IBC, is binding to be a tumor-suppressor protein DCN in aggressive IBC cell lines! Conceptually I find it difficult since they had to introduce DCN ectopically to counter their aggressive functions. The authors should clarify whether this a tumor-associated aberrant phenomenon. Since most of their functional data are derived with ectopic DCN over-expressing IBC cell lines, they should further specify where the DCN protein localized in these cells is. In their figures 3 e,f. the staining appears to be cytoplasmic.

They should devote a short section in the discussion to clarify this puzzle of how DCN in the cell membrane in low DCN-expressing IBC lines works as a tumor suppressor by binding to E-cadherin.

Response: Thank you for these great suggestions. We further confirmed our immunofluorescence findings by subcellular fractionation and immunoblotting. We isolated different fractions (cell membrane, cytoplasm, nuclei) of IBC cells. We demonstrated that DCN was detectable in the cell membrane and cytoplasmic, but not in the nuclear, fractions of IBC cells (**Supplementary Fig 12**). We used Histone H3 as the marker for the nuclear extract, vinculin as the marker for the cytoplasmic extract, and EGFR as a marker for the membrane extract.

As proposed in the model Fig 6e, because DCN protein expression was very low in IBC tumors and cells, we suspected that any association between endogenous DCN and E-cadherin (or EGFR) would not significantly affect ERK activation and IBC cell invasion. However, ectopic expression of DCN in IBC cells or treatment with DCN protein had significant effects on reducing expression of E-cadherin and suppressing EGFR-ERK signaling, which led to inhibition of invasion of IBC cells. We cannot exclude the possibility that stroma-derived DCN could also contribute to tumor suppressor function in IBC, because others have linked reduced levels of stromal DCN with more aggressive disease. Further characterization of the DCN-expressing cancer cells or stroma cell types, and cell-type-specific knockout of stromal DCN, may be critical to fully demonstrate the roles of cancer cell or stroma-derived DCN in IBC tumor biology. We have added discussion of this point in the revised Discussion section.

Reviewer 2 (Remarks to the Author):

Inflammatory breast cancer (IBC) is one of highly aggressive and metastatic form of breast cancer. It is essential not only to understand the underlying molecular mechanisms contributing to the aggressive nature of the IBC but also to develop novel and more effective therapeutic interventions for this aggressive form of breast cancer. Decorin (DCN) has been shown to be a tumour suppressor in different types of cancer including breast cancer. In this study, Hui and colleagues focused on IBC and report that 1) DCN is downregulated in tumors from patients with IBC, 2) overexpression of DCN in IBC cells is accompanied by decreased migration, invasion, and cancer stem cells in vitro and inhibited IBC tumor growth and metastasis in vivo, and 3) DCN function as a suppressor of tumour invasion and growth in IBC. They also have shown that these effects were mediated by reducing E-cadherin expression and inhibiting EGFR/ERK signaling pathways. Indeed, DCN binding to E-cadherin in IBC cells were found to accelerate its degradation through an autophagy-linked lysosomal pathway. However, DCN-mediated inhibition of E-Cadherin was found to be insufficient for the induction of EMT in IBC cells. On the basis of their data presented, the authors concluded that DCN induced inhibition of tumorigenesis and metastasis in IBC cells is mediated via downregulation of the E-cadherin/EGFR/ERK axis.

Overall, this is an interesting study and the results presented suggest a novel mechanism of IBC pathobiology and potential targets for therapeutic intervention for this type of breast cancer.

Response: We greatly appreciate the positive comments, thank you very much.

Reviewer 3 (Remarks to the Author):

The manuscript written by Xiaoding Hu et al. is a well planned work with interesting findings. The exhibited results are benefit in breast cancer research field. I think it is suitable to be accepted to submit in the journal.

Response: We are very grateful for the positive comments, thank you.

REVIEWERS' COMMENTS:

Reviewer #1 (Remarks to the Author):

The authors have satisfactorily addressed all the comments in the revised manuscript